# Development of Pleiotropic TrkB and 5-HT_4_ Receptor Ligands as Neuroprotective Agents

**DOI:** 10.3390/molecules29020515

**Published:** 2024-01-19

**Authors:** Mirjana Antonijevic, Despoina Charou, Audrey Davis, Thomas Curel, Maria Valcarcel, Isbaal Ramos, Patricia Villacé, Sylvie Claeysen, Patrick Dallemagne, Achille Gravanis, Ioannis Charalampopoulos, Christophe Rochais

**Affiliations:** 1Normandie University, Unicaen, Centre d’Etudes et de Recherche sur le Médicament de Normandie (CERMN), 14000 Caen, France; mirjana.antonijevic@unicaen.fr (M.A.); audrey.davis@unicaen.fr (A.D.); patrick.dallemagne@unicaen.fr (P.D.); 2Department of Pharmacology, Medical School, University of Crete, 70013 Heraklion, Greece; dcharou@gmail.com (D.C.); gravanis@med.uoc.gr (A.G.); charalampn@uoc.gr (I.C.); 3Institute of Molecular Biology & Biotechnology, Foundation of Research & Technology-Hellas, 70013 Heraklion, Greece; 4IGF, Univ Montpellier, CNRS, INSERM, 34000 Montpellier, France; thomas.curel@igf.cnrs.fr (T.C.); sylvie.claeysen@igf.cnrs.fr (S.C.); 5Innoprot S.L, 48160 Derio, Spain; mvalcarcel@innoprot.com (M.V.); iramos@zabala.es (I.R.); pvillace@innoprot.com (P.V.)

**Keywords:** neurodegeneration, TrkB receptor, 5-HT_4_ receptor, neuronal survival, neurite differentiation

## Abstract

One common event that is the most detrimental in neurodegenerative disorders, even though they have a complex pathogenesis, is the increased rate of neuronal death. Endogenous neurotrophins consist of the major neuroprotective factors, while brain-derived neurotrophic factor (BDNF) and its high-affinity tyrosine kinase receptor TrkB are described in a number of studies for their important neuronal effects. Normal function of this receptor is crucial for neuronal survival, differentiation, and synaptic function. However, studies have shown that besides direct activation, the TrkB receptor can be transactivated via GPCRs. It has been proven that activation of the 5-HT_4_ receptor and transactivation of the TrkB receptor have a positive influence on neuronal differentiation (total dendritic length, number of primary dendrites, and branching index). Because of that and based on the main structural characteristics of LM22A-4, a known activator of the TrkB receptor, and RS67333, a partial 5-HT_4_ receptor agonist, we have designed and synthesized a small data set of novel compounds with potential dual activities in order to not only prevent neuronal death, but also to induce neuronal differentiation in neurodegenerative disorders.

## 1. Introduction

The quest to find a cure for neurodegenerative disorders has been ongoing for years. The pathogenesis of these disorders is quite challenging since many signaling cascades are involved. We have witnessed that targeting only one cascade has not provided a cure so far, but only provided symptomatic treatment. However, one event is common and highlighted in these diseases—the death of neurons. The balance between the death of neurons and their survival is regulated by neurotrophins and their receptors: the p75 pan-neurotrophin receptor (p75*^NTR^*), a member of the tumor necrosis factor (TNF) receptor family, and the tropomyosin-related kinase (Trk) transmembrane tyrosine kinase receptors (TrkA, TrkB, TrkC) [1].

Among these receptors, it has been shown that the TrkB receptor has a crucial role in neuronal survival, axonal and dendritic growth, and plasticity, as well as in adult neurogenesis, making it the most important neurotrophin receptor in the central nervous system (CNS) [2,3,4,5]. The importance of the TrkB receptor is also proven by the fact that its signaling pathways (triggered by brain-derived neurotrophic factor (BDNF) or neurotrophin 4/5 (NT-4/5)~13 kDa large proteins) are impaired in many neurodegenerative diseases (like Alzheimer’s disease [6,7,8,9], Parkinson’s disease [10,11], and Huntington’s disease [12]). Restoration of these signaling pathways has been shown to have a protective effect in these conditions [13,14]. Due to their nature, BDNF and NT-4/5 do not penetrate the blood–brain barrier [15] and, once administrated, tend to cause a lot of side effects [16]. These pharmacological obstacles have raised the necessity of developing small molecules acting as TrkB receptor activators to resemble neurotrophin actions. 

One such small molecule is LM22A-4 [17]; however, this molecule does not possess a desirable physicochemical profile, since it is quite hydrophilic (chromatographic hydrophobicity index (CHI) at pH = 7.4 is CHI_7_._4_ = 13.8) [18]. Another issue concerning this molecule is the controversy about its properties in being a direct activator of the TrkB receptor [19,20]. However, even if this activation of the TrkB receptor via LM22A-4 [17,21,22,23] is indirect, it cannot be disregarded.

Interestingly, several cell functions are regulated thanks to the cross-communication of GPCRs with tyrosine kinase receptors [24]. One of the GPCRs that is influencing on the activity of the TrkB receptor is the serotonin 5-HT_4_ receptor [25]. Based on the fact that in most neurodegenerative diseases, the TrkB receptor is down-regulated [26], GRCR-mediated transactivation of the TrkB receptor could significantly improve its signaling and sustain the neuroprotective effects. It is of special note that a partial 5-HT_4_ receptor agonist, RS67333, was able to increase the expression of BDNF after 3 days of treatment, while a significant increment of BDNF was detected after 7 days of the regimen [25].

In light of these results, we have designed and synthesized a new data set of molecules in which we tried to keep the main structural characteristics of LM22A-4 and changed only its third position by introducing almost an entire RS67333 scaffold through its piperidine ring. The idea was to obtain compounds which will not only activate the TrkB receptor, but also act as partial 5-HT_4_ receptor agonists (Figure 1). This double activity could potentiate the neuroprotective effect of such compounds.

Indeed, in our previous work, we showed that by modifying the third position of LM22A-4, we can improve the TrkB activity [18]. Other studies have shown that a large variety of substitutions is allowed on the piperidine ring of RS67333 in order to obtain molecules that will possess an affinity towards the 5-HT_6_ receptor [27], antioxidant [28], or AChE inhibition [29] properties, while preserving good affinity and an agonistic profile towards the 5-HT_4_ receptor.

Based on these findings and taking in consideration the multivariant phenotype of neurodegenerative disorders, we aimed to create multi-target ligands capable of targeting the two aforementioned receptors, targeting the blockade of neuronal death and inducing the regeneration of neurons.

## 2. Results

### 2.1. Chemistry

The multistep synthesis started with the activation of 4-amino-5-chloro-2-methoxybenzoic acid (**1**) with CDI, followed by the synthesis of β-keto ester (**2**) that was further engaged in reaction of nucleophilic substitution with *N*-Boc-4-(iodomethyl)piperidine (**3**), followed by reactions of saponification and decarboxylation to afford intermediate **4** [30,31]. After the reaction, the intermediate 4 of -Boc deprotection was alkylated with *tert*-butyl-*N*-(2-bromoethyl)carbamate to afford intermediate **5** [28]. After another reaction of -Boc deprotection, we performed a reaction of peptidic coupling with the carboxylic acid (**6**), after which was the last step of amidation of two esters in order to obtain final product **8** (**ENT-C199**) in a 68% yield (Figure 1).

In order to assess the importance of the length and nature of the linker between two moieties, we decided to synthesize other analogues. As an outcome of one of these synthetic pathways, we obtained product **13** (**ENT-C232**) in a 68% yield, in which, instead of having an ethyl bridge, we introduced propyl one (Figure 2).

Besides the propyl bridge, we also synthesized a compound which possesses a diethyl ether bridge **19** (**ENT-C236**) in a 63% yield (Figure 3), which represents a true compromise between two scaffolds. The synthesis started with a reaction of protection of the free amine of 2-(2-aminoethoxy)ethanol (**14**) in order to obtain intermediate **15** (Figure 3). Intermediate **15** was used in the reaction of nucleophilic substitution with *p*-toluenesulfonyl chloride to afford corresponding tosylate **16** (Figure 3). Thanks to the generation of **16**, with which we performed the alkylation of **4** (after the -Boc deprotection) and obtained product **17** (Figure 3), we were able to obtain **19** (**ENT-C236**) in a 63% yield, after the last two steps that we previously described.

### 2.2. Biological Evaluation

#### 2.2.1. Cell Toxicity Assays in the NIH-3T3 TrkB Stable Transfected Cell Line

In order to evaluate if the novel compounds were able to activate the TrkB receptor, firstly the compounds were screened for their ability to reduce cell death by using the CellTox™ Green Cytotoxicity Assay [18]. This assay was used for testing the ability of the compounds to counter cell death caused by serum deprivation in NIH-3T3 TrkB cells that specifically express the TrkB receptor by measuring the levels of a dye that stains damaged DNA. The levels of cell death after BDNF or TrkB agonist treatments were normalized to cell death levels in the serum-free media (SFM) control, where maximal cell death is expected. The BDNF led to a reduction in cell death reduction of almost half (0.58) compared to the serum-free control levels. Similarly, treatment with compounds LM22A-4, ENT-C199 (Figure 2), and ENT-C232 (Figure 3) led to a significant reduction in toxicity, while ENT-C236 did not show any ability to rescue cells from cell death (Figure 3).

#### 2.2.2. TrkB Activation

The compounds exhibiting the most promising action at decreasing cell toxicity were tested for their ability to phosphorylate/activate the TrkB receptor and its downstream target, Akt kinase, in TrkB stably transfected NIH-3T3 cells [18]. In both cases, the phosphorylated and total protein levels were identified using Western blot analysis, as detailed in the relevant methods section. The relative phosphorylation levels were compared across the treatments after normalization compared to the control, using three independent experiments, while BDNF was used as a positive control. The phosphorylation levels in samples treated with compounds LM22A-4 and ENT-C232 were significantly higher than the SFM control, which was also repeated for the BDNF control in both cases. In contrast, ENT-C199 did not induce a significantly higher TrkB receptor or Akt phosphorylation compared to the SFM control (Figure 4).

#### 2.2.3. Evaluation of the 5-HT_4_R Binding Affinity

In order to evaluate if the newly synthesized molecules are indeed ligands for the 5-HT_4_ receptor, we conducted a binding study [32] on the target. The study was performed in duplicate in the absence or the presence of 10^−6^ or 10^−8^ M of the tested molecules and 0.5 nM [^3^H]-GR 113808. Nonspecific binding was evaluated in parallel in the presence of 30 µM serotonin. The results presented in Table 1 show that even though all the new molecules exert some affinity towards the 5-HT_4_ receptor, their affinity decreased with length of the introduced bridge, making ENT-C199 with an ethyl bridge between two moieties, the compound with the highest affinity. For ENT-C199, the affinity constant was calculated from five-point inhibition curves and expressed as Ki ± Sd (Table 2).

#### 2.2.4. Evaluation of the 5-HT_4_R Profile

Since we have proven that the newly designed molecules are 5-HT_4_ receptor ligands, we decided to determinate the pharmacological profile of the most potent one, as it was indicated by the binding assay results, ENT-C199. In this evaluation, two controls were used: serotonin (5-HT) and RS67333. As we observed, ENT-C199 behaves as partial agonist of the 5-HT_4_ receptor, transiently expressed in COS-7 cells (Figure 5).

#### 2.2.5. Efficacy Assay—Differentiation (Neurite Outgrowth Assay)

It is well-established knowledge that the BDNF/TrkB signaling pathway is important for dendritic and axonal growth [33,34,35] and, therefore, since the compounds described in our work were developed as potential TrkB receptor activators, we evaluated the compounds’ efficacy to induce neuronal differentiation. All the compounds were tested at three concentrations (10 nM, 100 nM, 1000 nM) on human SH-SY5 cells. We used three controls: control FBS (DMEMF12 with 10% FBS), control (cultured 3 days in DMEMF12 with 1% FBS, followed by 2 days in DMEMF12 FBS-free), and retinoic acid—RA (DMEMF12 with 1% FBS in the presence of RA 10 µM, followed by 2 days in DMEMF12 FBS-free) [18]. To investigate the role of neurite extension, the geometric patterns that were mainly employed in this study are the total neurite count, neurite total length, neurite average length, neurite critical value, neurite ramification index, and branch total point count. The results have been combined into one graph (Figure 6).

The results of this assay clearly show that besides the BDNF, ENT-C199 at 10 nM and 100 nM concentrations induced the differentiation of the cells (Figure 7). Interestingly, LM22A-4 was not capable of inducing neuronal differentiation. The compounds RS67333, ENT-C232, and ENT-C236 show a trend for inducing differentiation; however, their effect was not statistically significant.

### 2.3. In Silico Evaluation of the BBB Permeability

One of the crucial properties of the novel compounds that needed to be evaluated is their capability to penetrate the blood–brain barrier (BBB). In order to obtain more information, we performed an *in silico* study, in which we computed the physicochemical properties, as well as their potential to penetrate the BBB and be transferred into the central nervous system (CNS) using the SwissADME free web tool (http://www.swissadme.ch (accessed on 10 January 2024)) (Table 3).

## 3. Discussion

The necessity to develop molecules capable of preventing neurodegenerative processes fulfils an unmet medical need. Both referent molecules, a TrkB activator (LM22A-4) and a partial 5-HT_4_ receptor agonist (RS67333), have interesting profiles regarding the prevention of neuronal death. As the cross-communication between GPCRs and TrkB represents a promising field for the development of neurodegenerative treatments, we have tried to develop the first multi-target, TrkB/5-HT_4_ receptor ligands. 

Combining the LM22A-4 moiety and RS67333 scaffold, we obtained three different molecules: ENT-C199, ENT-C232, and ENT-C236.

Regarding the evaluation of the TrkB activity, while ENT-C199 and ENT-C232 were able to reduce induced cellular death of the NIH-3T3 cell line that expresses the TrkB receptor, treatment with ENT-C236, on the contrary, resulted in even higher levels of dead cells compared to the untreated control. Further assessment of the ability of the positive compounds to phosphorylate the TrkB receptor then showed that only ENT-C232 was able to increase the phosphorylation levels of TrkB and Akt, while ENT-C199 was not able to induce phosphorylation of either the TrkB receptor or Akt. Hence, these results prove that ENT-C232 acts as an activator of the TrkB receptor.

As we can see from the 5-HT_4_ receptor binding study, all three molecules indeed possess an affinity to bind to the receptor; however, that affinity decreases with an increase in chain length between the two moieties, meaning that the ENT-C199 with an ethyl chain between the two parts had the highest affinity for the receptor (measured affinity at 1 × 10^−8^ M was 24%, while at 1 × 10^−6^ M, it was 91%), while the ENT-C236 with the diethyl ether bridge had the lowest affinity (measured affinity at 1 × 10^−8^ M was 5%, while at 1 × 10^−6^ M, it was 55%).

Since we have proven that the newly designed molecules are 5-HT_4_ receptor ligands, we decided to determine the pharmacological profile of the most potent one, as it was indicated by the binding assay results, ENT-C199. In this evaluation, two controls were used: serotonin (5-HT) and RS67333. As we observed in three independent experiments, ENT-C199 behaved as a partial agonist of the 5-HT_4_ receptor, transiently expressed in COS-7 cells (Figure 5), which has shown us that the introduction of the LM22A-4 moiety on the piperidine ring of RS67333 does not influence the 5-HT_4_R activity.

After these first evaluations, we could see that ENT-C199 was behaving only like a partial 5-HT_4_ receptor agonist, while ENT-C232 had both the property of binding to the 5-HT_4_ receptor and inducing the phosphorylation of the TrkB receptor, making it the first multi-target TrkB/5-HT_4_ receptor ligand.

One of the influences that the molecules developed to prevent neurodegeneration must possess is the ability to induce neuronal differentiation, a property that we decided to evaluate for our new molecules. In the tested paradigm, we found that among the evaluated molecules, ENT-C199 was the most potent to induce neuronal differentiation. In this assay, even though BDNF as a TrkB ligand was capable of inducing the differentiation, LM22A-4, which is a small molecule TrkB activator, was not able to achieve a similar effect. This could be a consequence of the potential mechanism of action of LM22A-4, namely, it is claimed that LM22A-4 is not a direct ligand of the TrkB receptor and that its TrkB activity is actually caused by indirect activation trough some GPCR [20]. On the other side, we noticed the trend that RS67333 had in inducing neuronal differentiation, even though it was not statistically significant. This trend was also noticed in the results obtained after the treatment of cells with ENT-C232 and ENT-C236. Even if the three analogues are indeed 5-HT_4_ receptor ligands, ENT-C199 is the most potent and behaves as a partial 5-HT_4_ receptor agonist with an E_max_ lower than that of the RS67333. It is interesting to see that this molecule is the most powerful with regard to neuronal differentiation. We can hypothesize that this effect is due to the differentiating power of cAMP produced by the activation of the 5-HT_4_ receptor. Moreover, treating the SH-SY5Y cell line with retinoic acid increases the expression of the TrkB receptor [40]. If the affinity of ENT-C199 for this receptor has been established, its selectivity over other cell surface receptors, such as GPCRs, should be explored. Indeed, the cross-talk between GPCRs and the TrkB receptor is a known process [24], and a side activity might explain the differentiation of the cells.

Concerning the *in silico* prediction of the physicochemical and ADME properties, the important properties that we focused on were lipophilicity and BBB permeability. As we can see from the obtained results, the novel compounds indeed have increased lipophilicity; however, their predicted BBB permeation profile is not satisfying. Except for RS67333, for the remaining compounds, it was predicted that they do not cross BBB. This is because the prediction takes into consideration not only the passive permeation through the BBB (depending on the calculated values of LogP), but also if the molecules are substrates or non-substrates of the permeability glycoprotein (P-gP) [37]. As we know, P-gp are transporters that have a major role in protecting the CNS from xenobiotics, thanks to their capability to actively efflux foreign substances trough biological membranes [37]. That is why the predicted results have shown that the new molecules, despite being more lipophilic than LM22A-4, do not cross the BBB. However, more experiments are needed in order to definitively determine this property.

## 4. Materials and Methods

### 4.1. Chemistry

All the commercially available compounds were used without further purification. The melting points were determined on a Stuart melting point apparatus (SMP50). Analytical thin layer chromatography (TLC) was performed on silica gel 60 F254 on aluminium plates (Merck, Millipore, Burlington, MA, USA) and visualized with UV light (254 nm). Flash chromatography was conducted on a VWR SPOT II Essential instrument and the PURIFLASH C18-STD column, whose size and flow rate followed the manufacturer’s recommendations. The NMR spectra were recorded at 295 K, at 400 or 500 MHz (Bruker Avance III 400/500 MHz (Billerica, MA, USA)) for ^1^H NMR and at 100 or 126 MHz for ^13^C NMR in chloroform-*d*, methanol-*d*_4_, or DMSO-*d*_6_ with chemical shifts (δ) given in parts per million (ppm) relative to TMS as the internal standard. The following abbreviations are used when appropriate: d = doublet, t = triplet, q = quartet, m = multiplet, dd = doublet of peak splitting patterns, br = broad, s = singlet, doublet, dt = doublet of triplet. Coupling constants *J* are reported in hertz units (Hz). The infrared spectra (IR) were obtained on a PERKIN-ELMER FT-IR spectrometer (Codolet, France) and are reported in terms of frequency of absorption (cm^−1^) using KBr discs. The high-resolution mass spectra (HRMS) were obtained by electrospray (ESI, sampling cone 50 V, capillary 0.3 kV) on a Xevo G2-XS QTof WATERS mass spectrometer (Milford, MA, USA). The LC-MS (ESI) analyses were realized with a Waters Alliance 2695 as a separating module using the following gradients: A (95%)/B (5%) to A (5%)/B (95%) in 4.00 min. This ratio was held for 1.50 min before returning to the initial conditions in 0.50 min. The initial conditions were then maintained for 2.00 min (A = H_2_O, B = CH_3_CN; each containing HCOOH: 0.1%; column XBridge BEH C18 2.5 µm/3.0 × 50 mm; flow rate 0.8 mL/min). The MS were obtained on a SQ detector by positive ESI. The mass spectrum data are reported as *m*/*z* [18]. Full spectra are accessible in the Appendix A.

#### 4.1.1. General Procedure A for the Peptidic Coupling Reaction

A stirred solution of carboxylic acid (1.0 eq) and Et_3_N (1.5 eq) in dry THF (3.9 mL/mmol) was cooled at 0 °C. EDC (1.5 eq) was added, and the mixture was stirred at 0 °C for 5 min. HOBt (1.5 eq) was added, and the mixture was stirred at 0 °C for 15 min, after which amine (1.5 eq) was added, and the mixture was stirred at rt for 12–72 h. The crude was purified by chromatography on a silica gel column to afford the coupling product (32–83% yields) [18].

#### 4.1.2. General Procedure B for the Reaction of Amidation

To a solution of diester derivative (1.0 eq) in MeOH (10 mL/mmol), ethanolamine (8.0 eq) was added, and the mixture was stirred at 170 °C for 2–4 h. The crude was purified by chromatography on a silica gel column and/or by reverse phase flash chromatography to afford the expected diamides (52–63% yields) [18].

#### 4.1.3. General Procedure C for the Reaction of Alkylation

The corresponding amine was dissolved in dry DMF or can (10 mL/mmol), and 10 eq of K_2_CO_3_ was added. To the mixture, a corresponding alkylating reagent (1–1.4 eq) was added and the mixture was stirred at 80 °C or 110 °C for 2–10 h. The crude was purified by chromatography on a silica gel column and/or by reverse phase flash chromatography to afford the expected products (52–93% yields) [18].

*Ethyl-3-(4-amino-5-chloro-2-methoxyphenyl)-3-oxopropanoate* (**2**): 4-amino-5-chloro-2-methoxybenzoic acid (**1**) (1.20 g, 5.98 mmol), which was dissolved in dry THF (8.1 mL/mmol, 48.24 mL) and CDI (1.10 g, 6.58 mmol), was added portion-wise. The mixture was stirred at rt for 4 h, after which ethyl potassium malonate (1.22 g, 7.18 mmol) and magnesium chloride (684 mg, 7.48 mmol) were added portion-wise. The mixture was stirred at 40 °C for 24 h. Evaporation of the solvent gave a crude, which was diluted in water (100 mL), and the product was extracted with DCM (3 × 50 mL). The collected organic layers were washed with sat. NaHCO_3_, brine, and water, and dried over MgSO_4_. After evaporation of the solvent, the product was purified by chromatography on a silica gel column (Cyclohexane/EtOAc 80:20) to afford the product as a yellow solid (1.30 g, 80% yield).^1^H NMR (CDCl_3_-*d*, 399.7 MHz) δ 7.92 (s, 1H), 6.23 (s, 1H), 4.55 (s, 2H), 4.17 (q, *J* = 7.1 Hz, 2H), 3.87 (s, 2H), 3.82 (s, 3H), 1.24 (t, *J* = 7.1 Hz, 3H). ^13^C NMR (CDCl_3_-*d*, 100.4 MHz) δ 189.7, 168.7, 160.0, 148.8, 132.7, 117.5, 111.8, 97.2, 61.0, 55.6, 50.6, 14.3. The spectral and analytical data matched with the literature [31].

*Tert-butyl-4-(3-(4-amino-5-chloro-2-methoxyphenyl)-3-oxopropyl)piperidine-1-carboxylate* (**4**): To a solution of ethyl 3-(4-amino-5-chloro-2-methoxyphenyl)-3-oxopropanoate (**2**) (1.30 g, 4.78 mmol) in dry DMF (2.5 mL/mmol, 12 mL), *tert*-butyl-4-(iodomethyl)piperidine-1-carboxylate (**3**) (1.72 g, 5.31 mmol) and K_2_CO_3_ (1.32 g, 9.56 mmol) were added, and the mixture was stirred at rt for 48 h. The reaction mixture was diluted with water, and the product was extracted with EtOAc (3 × 50 mL). The organic layers were collected and washed with water and brine and dried over MgSO_4_. After evaporation of the solvent, a crude was obtained, which was used for the next step. The crude was dissolved in EtOH (170 mL, 24.29 mL/mmol). H_2_O (39 mL to give 0.6 M solution of KOH) and KOH (1.3 g, 23.2 mmol) were added, and the mixture was refluxed for 2.5 h. The EtOH was removed, and 100 mL of water were added. The product was extracted with EtOAc (3 × 45 mL), and the organic layers were collected and dried over MgSO_4_. After evaporation of the solvent, a crude was obtained, which was purified by chromatography on a silica gel column (DCM/EtOAc, gradient 100:0 to 80:20) to afford the product as a white solid (1.92 g, 74% yield). ^1^H NMR (CDCl_3_-*d*, 399.7 MHz) δ 7.79 (s, 1H), 6.26 (s, 1H), 4.46 (s, 2H), 4.09 (m, 2H), 3.85 (s, 3H), 2.91 (m, 2H), 2.69 (m, 2H), 1.67 (m, 2H), 1.60 (m, 2H), 1.45 (s, 9H), 1.45–1.39 (m, 1H), 1.10 (m, 2H). ^13^C NMR (CDCl_3_-*d*, 100.4 MHz) δ 198.9, 159.6, 155.0, 147.8, 132.4, 119.0, 111.4, 97.7, 79.3, 55.8 (2C), 44.4, 40.8, 35.9, 32.3 (2C), 31.3, 28.6 (3C). The spectral and analytical data matched with the literature [31].

*Tert-butyl-(2-(4-(3-(4-amino-5-chloro-2-methoxyphenyl)-3-oxopropyl)piperidin-1-yl)ethyl)carbamate* (**5**): The compound was prepared from *tert*-butyl-4-(3-(4-amino-5-chloro-2-methoxyphenyl)-3-oxopropyl)piperidine-1-carboxylate (**4**) (300 mg, 0.75 mmol) by first performing a reaction of the Boc deprotection in dry DCM (1.32 mL/mmol, 1.00 mL) and anhydrous TFA (0.50 mL, 6.49 mmol) at rt for 30 min. After evaporation of the solvents, the crude was used for the next step of alkylation with *tert*-butyl-(3-bromoethyl)carbamate (169 mg, 0.75 mmol) following general procedure C. The mixture was stirred at 110 °C for 6 h, after witch the reaction mixture was quenched with water (~50 mL) and extracted with EtOAc (3 × 25 mL). The collected organic layers were washed with brine and water and dried over MgSO_4_. The crude was purified by chromatography on a silica gel column (DCM/MeOH + 10%NH_4_OH, 95:5) to afford the product as a slightly yellow oil (174 mg, 52% yield). ^1^H NMR (CDCl_3_-*d*, 399.7 MHz) δ 7.79 (s, 1H), 6.26 (s, 1H), 3.85 (s, 3H), 3.26–3.21 (m, 2H), 2.92–2.87 (m, 4H), 2.50–2.42 (m, 2H), 2.04–1.91 (m, 2H), 1.70 (d, *J* = 9.8 Hz, 2H), 1.59 (q, *J* = 7.1 Hz, 2H), 1.45 (s, 9H), 1.34–1.24 (d, *J* = 15.0 Hz, 3H). ^13^C NMR (CDCl_3_-*d*, 100.4 MHz) 199.0, 159.4, 156.0, 147.7, 132.2, 118.9, 111.2, 97.5, 79.1, 57.5, 55.6, 53.8 (2C), 40.9, 37.4, 35.6, 32.2 (2C), 31.3, 28.5 (3C). The spectral and analytical data matched with the literature [28].

*1,3-dimethyl-5-[(2-{4-[3-(4-amino-5-chloro-2-methoxyphenyl)-3-oxopropyl]piperidin-1-yl}ethyl)carbamoyl]benzene-1,3-dicarboxylate carbamate* (**7**): The compound was prepared from 3,5-bis(methoxycarbonyl)benzoic acid (**6**) (70.0 mg, 0.29 mmol) and 1-(4-amino-5-chloro-2-methoxyphenyl)-3-[1-(2-aminoethyl)piperidin-4-yl]propan-1-one (149.8 mg, 0.44 mmol) according to general procedure A mixture and was stirred for 72 h. The crude was purified by chromatography on a silica gel column (DCM/MeOH + 10%NH_4_OH, gradient 100:0 to 80:20) to afford the product as a transparent oil (112.2 mg, 68% yield). ^1^H NMR (MeOD-*d*_4_, 399.7 MHz) δ 8.73 (t, *J* = 1.6 Hz, 1H), 8.70 (d, *J* = 1.6 Hz, 2H), 7.65 (s, 1H), 6.44 (s, 1H), 3.97 (s, 6H), 3.86 (s, 3H), 3.61 (t, *J* = 6.9 Hz, 2H), 3.14 (d, *J* = 11.6 Hz, 2H), 2.92 (t, *J* = 7.8 Hz, 2H), 2.73 (t, *J* = 6.7 Hz, 2H), 2.22 (t, *J* = 10.9 Hz, 2H), 1.80 (d, *J* = 11.2 Hz, 2H), 1.58 (q, *J* = 6.9 Hz, 2H), 1.41–1.32 (m, 2H), 1.32–1.21 (m, 1H). ^13^C NMR (MeOD-*d*_4_, 100.4 MHz) δ 200.9, 168.0, 166.8 (2C), 161.7, 151.7, 136.6, 133.9, 133.4 (2C), 132.9, 132.5 (2C), 117.8, 111.6, 98.1, 58.4, 56.1, 54.9 (2C), 53.1 (2C), 41.6, 37.8, 36.4, 32.6 (2C), 32.5. LC-MS *m*/*z* [M + H]^+^ 561.69/562.70. IR (KBr, cm^−1^) ν 2924, 1727, 1650, 1249, 740. HRMS/ESI: *m*/*z* calcd. for C_28_H_34_ClN_3_O_7_ [M]^+^ 560.2164, found 560.2170.

*N^1^-(2-{4-[3-(4-amino-5-chloro-2-methoxyphenyl)-3-oxopropyl]piperidin-1-yl}ethyl)-N^3^,N^5^-bis(2-hydroxyethyl)benzene-1,3,5-tricarboxamide* (**8**): The compound was prepared from 1,3-dimethyl-5-[(2-{4-[3-(4-amino-5-chloro-2-methoxyphenyl)-3-oxopropyl]piperidin-1-yl}ethyl)carbamoyl]benzene-1,3-dicarboxylate (7) (60.0 mg, 0.11 mmol) following general procedure B, and the mixture was stirred for 4 h. The crude was purified by chromatography on a silica gel column (DCM/MeOH + 10%NH_4_OH, gradient 100:0 to 80:20) to afford the product as a yellow oil (35.4 mg, 54% yield). ^1^H NMR (MeOD-*d*_4_, 399.7 MHz) δ 8.43 (s, 3H), 7.65 (s, 1H), 6.44 (s, 1H), 3.86 (s, 3H), 3.74 (t, *J* = 5.8 Hz, 4H), 3.58 (t, *J* = 6.8 Hz, 2H), 3.54 (t, *J* = 5.8 Hz, 4H), 3.04 (d, *J* = 11.6 Hz, 2H), 2.91 (t, *J* = 7.7 Hz 2H), 2.63 (t, *J* = 6.8 Hz, 2H), 2.09 (t, *J* = 10.9 Hz, 2H), 1.76 (d, *J* = 10.3 Hz, 2H), 1.56 (q, *J* = 6.9 Hz, 2H), 1.35–1.29 (m, 2H), 1.29–1.24 (m, 1H). ^13^C NMR (MeOD-*d*_4_, 100.4 MHz) δ 201.0, 168.9 (2C), 168.5, 161.7, 151.7, 136.6 (2C), 136.5, 132.9, 129.98 (2C), 117.9, 111.6, 98.1, 61.5 (2C), 58.5, 56.1, 55.0 (2C), 43.7 (2C), 41.6, 38.1, 36.7, 32.9 (2C), 32.7, 29.5. LC-MS *m*/*z* [M + H]^+^ 618.68/620.56. IR (KBr, cm^−1^) ν 3330, 1654, 1585, 1420, 1295, 1215, 709. HRMS/ESI: *m*/*z* calcd. for C_30_H_40_ClN_5_O_7_ [M]^+^ 618.2695, found 618.2708.

*Tert-butyl-(3-bromopropyl)carbamate* (**10**): (Boc)_2_O (4.80 g, 21.99 mmol) was dissolved in DCM (1.11 mL/mmol, 24.4 mL) and cooled at 0 °C. 2-Bromopropylamine (**9**) (5.34 g, 24.4 mmol) was added, followed by the drop-wise addition of Et_3_N (3.34 g (4.6 mL), 33 mmol). The mixture was allowed to warm to rt, and it was stirred for 48 h. The solvent was evaporated, and 75 mL of EtOAc was added. The solution was washed with sat. NH_4_Cl (2 × 25 mL), sat. NaHCO_3_ (2 × 25 mL), and brine (2 × 25 mL). The organic phase was dried over MgSO_4_, evaporated, and purified by chromatography on a silica gel column (Cyclohexane/DCM, gradient 100:0 to 25:75) to afford the product as a slightly yellow oil (4.49 g, 86% yield). ^1^H NMR (CDCl_3_-*d*, 399.7 MHz) δ 4.66 (s, 1H), 3.44 (t, *J* = 6.5 Hz, 2H), 3.27 (q, *J* = 6.5 Hz, 2H), 2.09–2.00 (m, 2H), 1.44 (s, 9H). ^13^C NMR (CDCl_3_-*d*, 100.4 MHz) δ 156.1, 79.6, 39.1, 32.8, 31.0, 28.5 (3C). The spectral and analytical data matched with the literature [41]. 

*Tert-butyl-(3-(4-(3-(4-amino-5-chloro-2-methoxyphenyl)-3-oxopropyl)piperidin-1-yl)propyl)carbamate* (**11**): The compound was prepared from *tert*-butyl-4-(3-(4-amino-5-chloro-2-methoxyphenyl)-3-oxopropyl)piperidine-1-carboxylate (**4**) (200 mg, 0.50 mmol) by first performing a reaction of the Boc deprotection in dry DCM (1.32 mL/mmol, 0.66 mL) and anhydrous TFA (0.33 mL, 4.32 mmol) at rt for 30 min. After evaporation of the solvents, the crude was used for the next step of alkylation with *tert*-butyl-(3-bromopropyl)carbamate (**10**) (165 mg, 0.70 mmol) following general procedure C. The mixture was stirred at 110 °C for 2 h, after which the reaction mixture was quenched with water (~50 mL) and extracted with EtOAc (3 × 25 mL). The collected organic layers were washed with brine and water and dried over MgSO_4_. The crude was purified by chromatography on a silica gel column (DCM/MeOH + 10%NH_4_OH, 95:5) to afford the product as a slightly yellow oil (155 mg, 68% yield). ^1^H NMR (CDCl_3_-*d*, 399.7 MHz) δ 7.78 (s, 1H), 6.26 (s, 1H), 5.46 (s, 1H), 4.49 (s, 2H), 3.84 (s, 3H), 3.19–3.15 (m, 2H), 3.11–3.04 (m, 2H), 2.92–2.85 (m, 2H), 2.56–2.50 (m, 2H), 2.12–2.05 (m, 2H), 1.76–1.72 (m, 4H), 1.64–1.57 (m, 2H), 1.42 (s, 9H), 1.38–1.33 (m, 3H). ^13^C NMR (CDCl_3_-*d*, 100.4 MHz) δ 198.8, 159.7, 156.3, 147.9, 132.3, 118.9, 111.4, 97.6, 79.1, 56.8, 55.8, 53.8 (2C), 40.7, 39.6, 35.0, 31.4 (2C), 30.9, 28.5 (3C), 26.0. LC-MS *m/z* [M + H]^+^ 455.82/456.81. IR (KBr, cm^−1^) ν 3347, 2929, 1691, 1622, 1586, 1252, 1214, 1175. HRMS/ESI: *m*/*z* calcd. for C_23_H_36_ClN_3_O_4_ [M]^+^ 454.2473, found 454.2469.

*1,3-dimethyl-5-[(2-{4-[3-(4-amino-5-chloro-2-methoxyphenyl)-3-oxopropyl]piperidin-1-yl}propyl)carbamoyl]benzene-1,3-dicarboxylate* (**12**): The compound was prepared from 3,5-bis(methoxycarbonyl)benzoic acid (**6**) (53.0 mg, 0.22 mmol) and 1-(4-amino-5-chloro-2-methoxyphenyl)-3-(1-(3-aminopropyl)piperidin-4-yl)propan-1-one (116.9 mg, 0.33 mmol) according to general procedure A, and the mixture was stirred for 48 h. The crude was purified by chromatography on a silica gel column (DCM/MeOH + 10%NH_4_OH, 95:05) to afford the product as a transparent oil (79.0 mg, 62% yield). ^1^H NMR (MeOD-*d*_4_, 399.7 MHz) δ 8.69 (t, *J* = 1.6 Hz, 1H), 8.63 (d, *J* = 1.6 Hz, 2H), 7.62 (s, 1H), 6.42 (s, 1H), 3.96 (s, 6H), 3.83 (s, 3H), 3.48–3.43 (m, 2H), 3.06–2.99 (m, 2H), 2.91–2.85 (m, 2H), 2.54–2.48 (m, 2H), 2.09–2.00 (m, 2H), 1.91–1.82 (m, 2H), 1.77–1.70 (m, 2H), 1.57–1.49 (m, 2H), 1.35–1.22 (m, 3H). ^13^C NMR (MeOD-*d*_4_, 100.4 MHz) δ 200.8, 167.8, 166.7 (2C), 161.6, 151.7, 136.8, 133.8, 133.2 (2C), 132.8, 132.4 (2C), 117.8, 111.6, 98.1, 57.7, 56.1, 54.9 (2C), 53.1 (2C), 41.6, 39.9, 36.6, 32.8 (2C), 32.6, 27.0. LC-MS *m*/*z* [M + H]^+^ 575.67/576.66. IR (KBr, cm^−1^) ν 3339, 2927, 1727, 1648, 1586, 1446, 1246, 1214, 740. HRMS/ESI: *m*/*z* calcd. for C_29_H_36_ClN_3_O_7_ [M]^+^ 574.2320, found 574.2321.

*N^1^-(3-(4-(3-(4-amino-5-chloro-2-methoxyphenyl)-3-oxopropyl)piperidin-1-yl)propyl)-N^3^,N^5^-bis(2-hydroxyethyl)benzene-1,3,5-tricarboxamide* (**13**): The compound was prepared from 1,3-dimethyl-5-[(2-{4-[3-(4-amino-5-chloro-2-methoxyphenyl)-3-oxopropyl]piperidin-1-yl}propyl)carbamoyl]benzene-1,3-dicarboxylate (**12**) (79 mg, 0.14 mmol) following general procedure B, and the mixture was stirred for 2 h. The crude was purified by reverse flash chromatography (H_2_O/ACN, gradient 100:0 to 75:25) to afford the product as a yellow oil (45 mg, 52% yield). ^1^H NMR (MeOD-*d*_4_, 399.7 MHz) δ 8.44 (t, *J* = 1.7 Hz, 1H), 8.41 (d, *J* = 1.7 Hz, 2H), 7.65 (s, 1H), 6.45 (s, 1H), 3.85 (s, 3H), 3.73 (t, *J* = 5.8 Hz, 4H), 3.53 (t, *J* = 5.8 Hz, 4H), 3.46 (t, *J* = 6.7 Hz, 2H), 3.06 (d, *J* = 11.1 Hz, 2H), 2.93–2.88 (m, 2H), 2.59–2.53 (m, 2H), 2.15–2.06 (m, 2H), 1.88 (p, *J* = 7.0 Hz, 2H), 1.80–1.73 (m, 2H), 1.58–1.50 (m, 2H), 1.34–1.23 (m, 3H). ^13^C NMR (MeOD-*d*_4_, 100.4 MHz) δ 200.9, 170.3, 168.8 (2C), 168.7, 161.7, 151.7, 136.7 (2C), 132.9, 130.0 (2C), 129.9, 117.8, 111.6, 98.1, 61.5 (2C), 57.5, 56.1, 54.8 (2C), 43.7 (2C), 41.6, 39.6, 36.4, 32.6 (2C), 32.5, 27.0. LC-MS *m*/*z* [M + H]^+^ 633.64/634.67. IR(KBr, cm^−1^) ν 3322, 2927, 1650, 1587, 1537, 1420. HRMS/ESI: *m*/*z* calcd. for C_31_H_42_ClN_5_O_7_ [M]^+^ 632.2851, found 632.2854.

*Tert-butyl-(2-(2-hydroxyethoxy)ethyl)carbamate* (**15**): 2-(2-aminoethoxy)ethanol (**14**) (2.0 g, 19.00 mmol) was dissolved in dry DCM (1.32 mL/mmol, 25 mL) and cooled at 0 °C. (Boc)_2_O (4.53 g, 20.76 mmol) was added, and the mixture was warmed to rt, and then stirred for 9 h. The solvent mixture was quenched with water and washed with brine (2 × 40 mL) and water (2 × 40 mL) and dried over MgSO_4_. Evaporation of solvent gave the product as a transparent oil (2.90 g, 74% yield). ^1^H NMR (CDCl_3_-*d*, 399.7 MHz) δ 5.07 (s, 1H), 3.75–3.70 (m, 2H), 3.58–3.52 (m, 4H), 3.34–3.28 (m, 2H), 1.43 (s, 9H). ^13^C NMR (CDCl_3_-*d*, 100.4 MHz) δ 156.2, 79.5, 72.3, 70.4, 61.8, 40.4, 28.5 (3C). The spectral and analytical data matched with the literature [42]. 

*2-(2-((tert-butoxycarbonyl)amino)ethoxy)ethyl 4-methylbenzenesulfonate* (**16**): *Tert*-butyl-(2-(2-hydroxyethoxy)ethyl)carbamate (**15**) (1.00 g, 4.87 mmol) was dissolved in dry DCM (10 mL/mmol,48 mL), and Et_3_N (1.97 g, 19.47 mmol) was added. The mixture was cooled at 0 °C and stirred for 5 min. p-toluenesulfonyl chloride (1.86 g, 9.76 mmol) was added, and the mixture was warmed to rt and stirred overnight. The solvent mixture was quenched with sat. NaHCO_3_, and the organic layer was collected and washed with brine (2 × 30 mL) and water (2 × 30 mL) and dried over MgSO_4_. The crude was purified by flash chromatography on a silica gel column (DCM/MeOH, gradient 100:0 to 98:02) to afford the product as a transparent oil (1.2 g, 69% yield). ^1^H NMR (CDCl_3_-*d*, 399.7 MHz) δ 7.80 (d, *J* = 8.4 Hz, 2H), 7.35 (dd, *J* = 7.8, 0.8 Hz, 2H), 4.81 (s, 1H), 4.19–4.12 (m, 2H), 3.67–3.57 (m, 2H), 3.48–3.42 (m, 2H), 3.24 (t, *J* = 5.4 Hz, 2H), 2.45 (s, 3H), 1.44 (s, 9H). ^13^C NMR (CDCl_3_-*d*, 100.4 MHz) δ 156.0, 145.1, 133.1, 129.9 (2C), 128.1 (2C), 79.5, 70.5, 69.2, 68.5, 40.3, 28.5 (3C), 21.8. The spectral and analytical data matched with the literature [43]. 

*Tert-butyl-(2-(2-(4-(3-(4-amino-5-chloro-2-methoxyphenyl)-3-oxopropyl)piperidin-1-yl)ethoxy)ethyl)carbamate* (**17**): The compound was prepared from *tert*-butyl-4-(3-(4-amino-5-chloro-2-methoxyphenyl)-3-oxopropyl)piperidine-1-carboxylate (**4**) (150.0 mg, 0.38 mmol) by first performing a reaction of the Boc deprotection in dry DCM (1.32 mL/mmol, 0.5 mL) and anhydrous TFA (0.25 mL, 3.29 mmol) at rt for 30 min. After evaporation of the solvents, the crude was used for the next step of alkylation with 2-(2-((*tert*-butoxycarbonyl)amino)ethoxy)ethyl-4-methylbenzenesulfonate (**16**) (163.0 mg, 0.45 mmol) in ACN following general procedure C. The mixture was stirred at 80 °C for 10 h, after which the reaction mixture was quenched with water (~50 mL) and extracted with EtOAc (3 × 25 mL). The collected organic layers were washed with brine and water and dried over MgSO_4_. The crude was purified by chromatography on a silica gel column (DCM/MeOH + 10%NH_4_OH, 95:5) to afford the product as a slightly yellow oil (142.8 mg, 93% yield). ^1^H NMR (MeOD-*d*_4_, 399.7 MHz) 7.65 (s, 1H), 6.45 (s, 1H), 3.86 (s, 3H), 3.59 (t, *J* = 5.5 Hz, 2H), 3.46 (t, *J* = 5.5 Hz, 2H), 3.21 (t, *J* = 5.5 Hz, 2H), 2.99 (d, *J* = 11.1 Hz, 2H), 2.93–2.88 (m, 2H), 2.58 (t, *J* = 5.5 Hz, 2H), 2.06 (t, *J* = 10.4 Hz, 2H), 1.76–1.68 (m, 2H), 1.60–1.51 (m, 2H), 1.43 (s, 9H), 1.33–1.25 (m, 3H). ^13^C NMR (MeOD-*d*_4_, 100.4 MHz) δ 200.9, 161.6, 158.5, 151.7, 132.9, 117.9, 111.6, 98.1, 80.0, 70.8, 68.9, 59.1, 56.1, 55.2(2C), 41.6, 41.3, 36.6, 32.7(2C), 32.6, 28.8(3C). LC-MS *m/z* [M + H]^+^ 485.80/486.81. IR (KBr, cm^−1^) ν 3349, 2927, 1705, 1621, 1586, 1453, 1251, 1214, 1174. HRMS/ESI: *m*/*z* calcd. for C_24_H_38_ClN_3_O_5_ [M]^+^ 484.2578, found 484.2579.

*1,3-dimethyl-5-[(2(2-{4-[3-(4-amino-5-chloro-2-methoxyphenyl)-3-oxopropyl]piperidin-1-yl}ethoxy)ethyl)carbamoyl]benzene-1,3-dicarboxylate* (**18**): The compound was prepared from 3,5-bis(methoxycarbonyl)benzoic acid (**6**) (55.0 mg, 0.23 mmol) and 1-(4-amino-5-chloro-2-methoxyphenyl)-3-(1-(2-(2-aminoethoxy)ethyl)piperidin-4-yl)propan-1-one (133.0 mg, 0.35 mmol) according to general procedure A. The mixture was stirred for 48 h. The crude was purified by chromatography on a silica gel column (DCM/MeOH + 10%NH_4_OH, 95:5) to afford the product as a transparent oil (75.0 mg, 54% yield). ^1^H NMR (MeOD-*d*_4_, 399.7 MHz) δ 8.60–8.59 (m, 1H), 8.58 (d, *J* = 1.5 Hz, 2H), 7.53 (s, 1H), 6.33 (s, 1H), 3.84 (s, 6H), 3.72 (s, 3H), 3.57–3.42 (m, 6H), 2.92–2.80 (m, 2H), 2.78–2.62 (m, 2H), 2.52–2.44 (m, 2H), 1.96–1.89 (m, 2H), 1.54–1.46 (m, 2H), 1.42–1.33 (m, 2H), 1.14–1.04 (m, 3H). ^13^C NMR (MeOD-*d*_4_, 100.4 MHz) δ 200.9, 167.9, 166.7 (2C), 161.6, 151.7, 136.8, 133.8, 133.4 (2C), 132.8, 132.4 (2C), 117.8, 111.6, 98.1, 70.3, 69.1, 58.9, 56.1, 55.2 (2C), 53.1 (2C), 41.6, 41.1, 36.5, 32.7 (2C), 32.6. LC-MS *m*/*z* [M + H]^+^ 605.60/606.61. IR (KBr, cm^−1^) ν 3346, 2927, 1729, 1587, 1549, 1522, 1449, 1249, 1215, 742. HRMS/ESI: *m*/*z* calcd. for C_30_H_38_ClN_3_O_8_ [M]^+^ 604.2426, found 604.2427.

*N^1^-(2-(2-(4-(3-(4-amino-5-chloro-2-methoxyphenyl)-3-oxopropyl)piperidin-1-yl)ethoxy)ethyl)-N^3^,N^5^-bis(2-hydroxyethyl)benzene-1,3,5-tricarboxamide* (**19**): The compound was prepared from 1,3-dimethyl-5-[(2(2-{4-[3-(4-amino-5-chloro-2-methoxyphenyl)-3-oxopropyl]piperidin-1-yl}ethoxy)ethyl)carbamoyl]benzene-1,3-dicarboxylate (**18**) (75.0 mg, 0.12 mmol) following general procedure B. The mixture was stirred for 2 h. The crude was purified by reverse flash chromatography (H_2_O/ACN, gradient 100:0 to 75:25) to afford the product as a slightly yellow oil (51.3 mg, 63% yield). ^1^H NMR (MeOD-*d*_4_, 399.7 MHz) δ 8.37–8.35 (m, 1H), 8.35 (d, *J* = 1.7 Hz, 2H), 7.55 (s, 1H), 6.34 (s, 1H), 3.74 (s, 3H), 3.62 (t, *J* = 5.8 Hz, 4H), 3.57–3.48 (m, 6H), 3.42 (t, *J* = 5.8 Hz, 4H), 2.87 (d, *J* = 11.7 Hz, 2H), 2.79–2.72 (m, 2H), 2.50 (t, *J* = 5.4 Hz, 2H), 1.93 (t, *J* = 11.3 Hz, 2H), 1.53 (d, *J* = 12.1 Hz, 2H), 1.39 (q, *J* = 7.0 Hz, 2H), 1.17–1.00 (m, 3H). ^13^C NMR (MeOD-*d*_4_, 100.4 MHz) 201.1, 168.8 (2C), 168.7, 161.7, 151.7, 136.6 (3C), 132.9, 130.1 (2C), 130.0, 117.9, 111.6, 98.1, 70.3, 68.9, 61.5 (2C), 59.0, 56.1, 55.1 (2C), 43.7 (2C), 41.6, 41.0, 36.5, 32.7, 32.6 (2C). LC-MS *m*/*z* [M + H]^+^ 663.57/664.53. IR (KBr, cm^−1^) ν 3331, 1652, 1588, 1544, 1420, 1215. HRMS/ESI: *m*/*z* calcd. for C_32_H_44_ClN_5_O_8_ [M]^+^ 662.2957, found 662.2964.

### 4.2. TrkB Phosphorylation and Cell Survival

#### 4.2.1. Cell Culture

NIH-3T3 cells, both naive and stably transfected with the TrkB receptor, were kindly provided by Dr. Carlos F. Ibanez (Karolinska Institute and Peking University). The NIH-3T3 cells stably transfected to express the TrkB receptor were used to test the ability of the compounds to reduce cell toxicity after serum deprivation and induce TrkB phosphorylation. The cells were cultured in high-glucose DMEM medium supplemented with 10% fetal bovine serum (FBS), 100 units/mL penicillin, and 100 µg/mL streptomycin at 37 °C in a humidified 5% CO_2_ atmosphere [18].

#### 4.2.2. Western Blot

The cells were treated with BDNF at 500 ng/mL (positive control) or compounds at 1 μM for 20 min. After the compound treatment, the cells were plated in 12-well plates at 100,000 cells/well and were serum deprived for 6 h on the following day. 

The cells were lysed on ice for 10 min in Pierce IP lysis buffer (Thermo Fisher Scientific, Waltham, MA, USA), using a phospho-protease inhibitor cocktail by Millipore. A loading buffer (5× Laemni) was added to 25 μg from each protein sample, followed by incubation at 95 °C for 5 min and SDS-PAGE electrophoresis. The proteins were transferred to a nitrocellulose membrane for 2 h at 350 mA. The membrane was blocked with 5% bovine serum albumin (BSA) for 1 h at room temperature and incubated with the primary antibodies in blocking solutions at 4 °C overnight before detection with HRP-conjugated secondary antibodies. Chemiluminescence was detected with ECL solution. 

Primary antibodies used: Phospho-TrkB (Tyr816) Millipore # ABN1381 (Merck Millipore, Burlington, MA, USA), Anti-TrkB Abcam #ab33655 (Abcam plc., Cambridge, UK), Phospho-Akt (Ser473) CST #9271, Akt CST #9272 (Cell Signaling Technology, Danvers, MA, USA), GADPH Sigma # G8795 (Sigma-Aldrich, St. Louis, MO, USA) [18].

#### 4.2.3. Celltox Green Cytotoxicity Assay

The NIH-3T3 TrkB stably transfected cells were plated in 96-wells plates at 10,000 cells/well. The cells were subjected to serum starvation for 24 h prior to 24 h treatment with BDNF at 500 ng/mL or compounds at 1 μM. CellTox™ green dye (2000×, Promega (Promega Corporation, Maddison, WI, USA)) and Hoechst 33,342 solution (10,000×) were used to visualize the dead and total cells, respectively. Images were captured using an Olympus microscope and analyzed using ImageJ software (https://imagej.nih.gov/ij/ (accessed on 20 February 2019)) [18].

### 4.3. Binding Evaluation of Drugs on Human 5-HT_4_R

The method was validated from saturation studies: six concentrations of [^3^H]-GR113808 were used to give the final concentrations of 0.0625–2 nM, and nonspecific binding of [^3^H]-GR113808 was defined in the presence of 30 μM serotonin to determine the K_d_ and the B_max_. The competition studies used membrane preparations made from proprietary stable recombinant cell lines expressing the 5-HT_4(b)_ receptor to ensure high-level of GPCR surface expression (HTS110M, Millipore). The membranes (2.5 µg protein) were incubated in duplicate at 25 °C for 60 min in the absence or the presence of 10^−6^ or 10^−8^ M of each drug and 0.5 nM [^3^H]-GR113808 (VT 240, ViTrax) in 25 mM Tris buffer (pH 7.4, 25 °C). At the end of the incubation, the homogenates were filtered through Whatman GF/C filters (Alpha Biotech, Glasgow, Scotland) pre-soaked with 0.5% polyethyleneimine using a Brandel cell harvester. The filters were subsequently washed three times with 4 mL of ice-cold 25 mM Tris bufer (pH 7.4, 4 °C). Non-specific binding was evaluated in parallel in the presence of 30 μM serotonin. For some of these compounds, the affinity constants were calculated from five-point inhibition curves using the EBDA-Ligand software and expressed as Ki ± Sd [30].

### 4.4. Determination of cAMP Production

COS-7 cells were purchased from ATCC (ATCC CRL-1651; LGC STANDARTS, Molsheim, France). They were grown in Dulbecco’s modified Eagle medium (DMEM) supplemented with 10% dialyzed fetal calf serum (dFCS) and antibiotics. The cells were transiently transfected with the plasmid encoding HA-tagged human 5-HT_4_ receptor, then seeded in 96-wells plate (35,000 cells/well). Approximately 24 hrs after transfection, the cells were washed once with 200 µL of HBS (20 mM HEPES; 150 mM NaCl; 4.2 mM KCl; 0.9 mM CaCl_2_; 0.5 mM MgCl_2_; 0.1% glucose; 0.1% BSA) and, after HBS removal, exposed to the indicated concentrations of 5-HT_4_R ligands in the presence of 0.1 mM of the phosphodiesterase inhibitor RO-20-1724, at 37 °C in 100 µL of HBS. After 10 min, the cells were then lysed by the addition of the same volume of Triton-X100 (0.1%) during 30 min at 37 °C. Quantification of the cAMP production was performed by HTRF by using the cAMP—HTRF Gs Dynamic kit (Perkin Elmer, Codolet, France) according to the manufacturer’s instructions.

### 4.5. Efficacy Assay—Differentiation (Neurite Outgrowth Assay)

Human SH-SY5 cells were obtained from the HPA collection. The cells were cultured in endotoxin-free DMEM-F12 medium supplemented with 10% fetal bovine serum (FBS) and 100 units/mL penicillin and 100 µg/mL streptomycin (all the tissue culture reagents were purchased from Sigma-Aldrich, Madrid, Spain). The cultures were maintained at 37 °C in a humidified atmosphere with 5% CO_2_ and passaged every 3–4 days by trypsinization and were seeded into corning flasks at 2 × 10^6^ cells/flask [18]. 

#### Experimental Procedure

For the experimental assays, the cells were plated in 96-wells plate, previously coated with Matrigel matrix (Corning Cat. 354234) 1:15, with a number of 1250 cells per well in complete medium for 24 h at 37 °C in a humidified 5% CO_2_ atmosphere. At 24 h, the cells were treated with retinoic acid (RA) 10 µM for 3 days in the presence of DMEMF-12 plus 1% FBS. Three days later, the RA was discarded, and BDNF 20 ng/mL and the compounds (10 nM, 100 nM, 1000 nM) were added in DMEM-F12 serum-free for 2 days. Different control cultures were used in this assay: some cells were cultured with DMEM-F12 plus 10% FBS for the entire assay (five days—undifferentiated cells); other cells were cultured 3 days in DMEM-F12 plus 1% FBS, followed by 2 days in DMEM-F12 FBS-free; other control cells were cultured in DMEM-F12 plus 1% FBS in the presence of RA 10 µM, followed by 2 days in DMEM-F12 FBS-free. 

After 5 days, the cells were fixed with PFA for 15 min. After the fixation step, the samples were washed three times with PBS and permeabilized with PBS + 0.3% triton for 10 min. The samples were then blocked with PBS + bovine serum albumin (BSA) for 30 min, and, finally, an anti-tubulin III antibody was added at 1/1000 in PBS + 0.5% BSA for 60 min at room temperature. After three washing steps, the secondary antibodies, Alexa 633, were added at 1/100 for 60 min in order to react against the primary antibody. The samples were then washed three times and measured in a Pathway 855 automated fluorescent microscope. To investigate the role of neurite extension, geometric patterns were mainly employed in this study: neurite total count, neurite total length, neurite average length, neurite critical value, neurite ramification index, and branch total point count [18].

## 5. Conclusions

The development of selective TrkB receptor ligands by itself represents quite a challenge since the existing activators are large proteins, and the reported small molecules seem to activate the receptor through indirect ways. This quest becomes even more challenging when you want to develop a molecule that simultaneously targets the TrkB receptor and a GPCR. We decided specifically to merge these two families of receptors, not only because of the existing cross-communication, but because the interaction between the TrkB receptor and GPCRs represents a unique field from the aspect of medicinal chemistry, as well. To our knowledge, there are no multi-target ligands that have the TrkB receptor and a GPCR as a focus of action. In our quest, we used the known activator of the TrkB receptor and decided to merge it with a partial 5-HT_4_ receptor agonist in order to obtain molecules capable of not only preventing neuronal death, but also inducing the regeneration and development of already damaged ones. We could see that in our small study it was challenging to develop such molecules. Nevertheless, we were able to obtain one molecule, ENT-C232, which represents one of its kind—a multi-target TrkB/5-HT_4_ receptor ligand. However, we also raised a several questions regarding the cross-communication between the TrkB receptor and GPCRs: is the LM22A-4 indeed causing its TrkB effect through a GPCR? If so, which one? Which GPCR is crucial in the cross-communication with the TrkB receptor that is influencing neuronal differentiation? Since we know that ENT-C232 is able to activate the TrkB receptor and bind at the 5-HT_4_ receptor, it will be of great interest to see how this molecule will behave in more complex assays, such as protection of neuronal stem cells intoxicated with Aβ oligomers, or protection of Aβ-induced synapse degeneration of primary hippocampal neurons. Even though this is a small data set of molecules, our work represents a starting point in deciphering how the interaction of GPCRs and TrkB receptor can be used to prevent neurodegeneration, which would give hope that, perhaps, the moment for finding the cure is not that far away. 

## Data Availability

Data are contained within the article and Appendix A.

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
