# Peer review of "Development of Pleiotropic TrkB and 5-HT4 Receptor Ligands as Neuroprotective Agents"

_molecules, 2024, doi:10.3390/molecules29020515_

Round 1

Reviewer 1 Report

Comments and Suggestions for Authors

The manuscript "Development of pleiotropic TrkB/5-HT4 receptors ligands as neuroprotective agents" by Mirjana Antonijevic et al. presents the design and biochemical evaluation of 3 compounds as neuroprotective agents. The compounds are based on two known bioactive compounds, an activator of the tropomyosin-related kinase receptor TrkB, and a partial agonist of the serotonin receptor 5-HT4, based on the hypothesis that their potential dual activity could mimic the activity of natural neurotrophic factors, such as BDNF. Biochemical evaluation in human neuroblastoma cells revealed one compound (ENT-C199) with comparable differentiation activity with that of the BDNF, albeit not an activator of TrkB or Akt, at least to a high extent. Still, ENT-C199 was shown to be a partial agonist of the 5-HT4 receptor.

Overall the manuscript is very well-written and the results are presented in a clear and concise way, although the latter can be further improved. The chemistry is solid and the biochemical results are presented with appropriate statistical analysis. Similarly, the discussion is concise and to the point, but can be further improved too. Despite the small number of compounds synthesized and tested, I endorse publication of this work in Molecules as an exemplary and comprehensive medicinal chemistry work. Clear hypothesis, testing and solid investigation.

Below are some minor points meant to improve presentation of the results:

1. The x-axis labels of the top panels in Fig. 4 are missing. Quality of the graphs can be improved.

2. A more comprehensive description of the results obtained from the phosphorylation assays (l. 150-155) can be helpful.

3. Induction of neuronal differentiation by ENT-C236 seems to be within the statistical significance of ENT-C199 compared with BDNF (Graph 1), albeit of lower activity. Still, can the authors comment on the lower differentiation activity of all compounds at 1 μM?

4. Discussion on the hypothesis about another potential receptor that could be in play during differentiation can be improved. The levels of TrkB/Akt activation in conjunction with the binding affinity for the 5-HT4 receptor should be stated for each compound, so as to be clearly correlated with the corresponding neuronal differentiation effect and be compared with BDNF.

5. It would be helpful if the authors could present the chemical structure of compounds in the same page as their 1H and 13C NMR spectra (Supplementary Material).

Author Response

Reviewer #1: The manuscript "Development of pleiotropic TrkB/5-HT4 receptors ligands as neuroprotective agents" by Mirjana Antonijevic et al. presents the design and biochemical evaluation of 3 compounds as neuroprotective agents. The compounds are based on two known bioactive compounds, an activator of the tropomyosin-related kinase receptor TrkB, and a partial agonist of the serotonin receptor 5-HT4, based on the hypothesis that their potential dual activity could mimic the activity of natural neurotrophic factors, such as BDNF. Biochemical evaluation in human neuroblastoma cells revealed one compound (ENT-C199) with comparable differentiation activity with that of the BDNF, albeit not an activator of TrkB or Akt, at least to a high extent. Still, ENT-C199 was shown to be a partial agonist of the 5-HT4 receptor.

Overall the manuscript is very well-written and the results are presented in a clear and concise way, although the latter can be further improved. The chemistry is solid and the biochemical results are presented with appropriate statistical analysis. Similarly, the discussion is concise and to the point, but can be further improved too. Despite the small number of compounds synthesized and tested, I endorse publication of this work in Molecules as an exemplary and comprehensive medicinal chemistry work. Clear hypothesis, testing and solid investigation.

Below are some minor points meant to improve presentation of the results:

  1. The x-axis labels of the top panels in Fig. 4 are missing. Quality of the graphs can be improved.

We thank the reviewer for their helpful comments in improving our visual representations. We have now corrected the graph quality as discussed and added a label on the x axis (condition).

  1. A more comprehensive description of the results obtained from the phosphorylation assays (l. 150-155) can be helpful.

The relevant section has now been expanded as per suggestion to include an extended description of the western blot analysis and the relevant results.

  1. Induction of neuronal differentiation by ENT-C236 seems to be within the statistical significance of ENT-C199 compared with BDNF (Graph 1), albeit of lower activity. Still, can the authors comment on the lower differentiation activity of all compounds at 1 μM?

We thank the reviewer for raising an interesting point of discussion at this section. While initial experiments had indeed suggested ENT-C236 may have shown a trend for increased differentiation, the statistical analysis did not confirm this result to be significant compared to the control, as opposed to ENT-C199. This stems from the variance, with lower values being closer to the control, compared to ENT-C199, which is consistently higher.

Regarding lower differentiation activity at higher concentrations, desentitization of the 5-HT4 receptor may be responsible of the effect of all compounds at 1 μM quoting (Ansanay H et al. Mol Pharmacol. 1992 Nov;42(5):808-16. PMID: 1331763) and the bell-shaped curve which is classic for 5HT4 receptor agonists in functional tests. Note that the activity of the compounds at 1µM is inversely correlated with their binding affinity for 5-HT4R.

  1. Discussion on the hypothesis about another potential receptor that could be in play during differentiation can be improved. The levels of TrkB/Akt activation in conjunction with the binding affinity for the 5-HT4 receptor should be stated for each compound, so as to be clearly correlated with the corresponding neuronal differentiation effect and be compared with BDNF.

The discussion on this hypothesis has been reformulated according to the reviewer comments.

  1. It would be helpful if the authors could present the chemical structure of compounds in the same page as their 1H and 13C NMR spectra (Supplementary Material).

The chemical structure is presented beneath the title of each compound followed by its HRMS, 1H and 13C spectrums in the suppplementary section.

Reviewer 2 Report

Comments and Suggestions for Authors

The author designed and synthesized a novel compounds with potential dual activities on both TrkB and 5-HT4 receptors. It is very interesting paper in this field. The data supported their insights in this manuscript. The author should address my concern before pulication. The antagonists of both receptors should be used to evaludate the sepecificity of the new compound in the assays.

Comments on the Quality of English Language

I suggest the author should polish their presentation.

Author Response

Reviewer #2: The author designed and synthesized a novel compounds with potential dual activities on both TrkB and 5-HT4 receptors. It is very interesting paper in this field. The data supported their insights in this manuscript. The author should address my concern before pulication. The antagonists of both receptors should be used to evaludate the sepecificity of the new compound in the assays. I suggest the author should polish their presentation.

We thank the reviewer for their positive criticism on our work. Regarding the presentation of part of our work, we have now improved the main plots. We appreciate the reviewer’s input on studying the receptor specificity of the compounds and this is indeed a point we would like to thoroughly address in a follow up work, which it would demand pharmacokinetic/pharmacodynamic studies on both receptors systems. However, in the present work we decided not to include this line of experiments, since this is a time consuming experimentation and probably out of the scope of this manuscript. In this first study, we aimed to introduce a quick assessment of compounds that have pleiotropic neuroprotective action, using the TrkB/5-HT4 receptor pathways because of their significant involvement in these mechanisms. Therefore, we focused our efforts to identify the most effective compounds, as well as to clarify their mechanism of action on the receptor and the downstream signaling cascade. Of course, we are in full agreement with the reviewer’s opinion that a more extensive and detailed analysis is necessary in order to assess the suitability of these agents as putative drugs in the future, including in vivo studies in appropriate animal models of the disease.

Reviewer 3 Report

Comments and Suggestions for Authors

The manuscript entitled “Development of pleiotropic TrkB/5-HT4 receptors ligands as 2 neuroprotective agents” bears a small series of newly synthesized compounds able to bind simultaneously TrkB 24 and 5-HT4 receptors. This dual mechanism of action could be useful to prevent neuronal death and induce regeneration of neurons in neurodegenerative disorders. The new molecules probably possess an increased lipophilicity respect to the parent compound LM22A-4, which gives the compounds the ability to permeate the BBB.

The manuscript is well written and arranged. It provides a useful contribution to this area of research but, requires major revision before being published on Molecules. Requests are available below.

General comments

My major criticism concerns the choice and use of these cell lines (NIH 3T3 are mouse embryonic fibroblast cells, SH-SY5Y are human neuroblastoma cells). Having to study the neuroprotective effects of some compounds, I would study them on neuronal or microglial cell lines. I understand that this study is the starting point for a more in-depth one, but too many aspects remain to be clarified, first of all the ability of these compounds to cross the blood-brain barrier. I therefore propose to the authors to enrich the manuscript at least with studies on the permeability of the blood-brain barrier.

Minor revisions 

The manuscript includes some typing errors and it should be re-checked.

Page 2, line 27, the sentence “the light of these results, we have “should be deleted.

Page 6, line 145, compound LM22A-4 should be deleted in the caption of figure 3 since it is not included in the figure.

Page 7, line 172, “Ki” should be written as “Ki”. Please uniform the manuscript accordingly.

Page 8, table 21, the concentration of compounds tested should be 1.10-8 and 1.10-6 M, please change the table header accordingly.

Page 8, line 181, the sentence “behaves a partial agonist” should be changed in “behaves as partial agonist”.

Page 11, line 268, the question mark should be deleted.

Page 17, line 558, “0.0625–2vnM” should be changed in “0.0625–2 nM”.

Page 18, line 590, “2x106” should be changed in “2x106”.

Comments on the Quality of English Language

Overall the manuscript is well written and good English has been used, but some typos and careless errors need to be corrected.

Author Response

Reviewer #3: The manuscript entitled “Development of pleiotropic TrkB/5-HT4 receptors ligands as 2 neuroprotective agents” bears a small series of newly synthesized compounds able to bind simultaneously TrkB 24 and 5-HT4 receptors. This dual mechanism of action could be useful to prevent neuronal death and induce regeneration of neurons in neurodegenerative disorders. The new molecules probably possess an increased lipophilicity respect to the parent compound LM22A-4, which gives the compounds the ability to permeate the BBB.

The manuscript is well written and arranged. It provides a useful contribution to this area of research but, requires major revision before being published on Molecules. Requests are available below.

General comments

My major criticism concerns the choice and use of these cell lines (NIH 3T3 are mouse embryonic fibroblast cells, SH-SY5Y are human neuroblastoma cells). Having to study the neuroprotective effects of some compounds, I would study them on neuronal or microglial cell lines. I understand that this study is the starting point for a more in-depth one, but too many aspects remain to be clarified, first of all the ability of these compounds to cross the blood-brain barrier. I therefore propose to the authors to enrich the manuscript at least with studies on the permeability of the blood-brain barrier.

We appreciate the reviewer’s positive take on our work and would like to clarify certain points of the scope of this manuscript we may have failed to fully illustrate.

NIH 3T3 (and COS-7 cells...) are used here to characterize the profile of the compounds regarding both targets.

First of all, we used NIH-3T3 cells as an established system for fast and efficient screening that has been previously used successfully for this purpose. Naïve and TrkB-transfected NIH-3T3 cells were kindly provided from our collaborator, Dr Carlos F. Ibanez. This cell line of fibroblasts provides a selective cell cassette system to screen TrkB-dependent signaling, since the comparison between TrkB-stably transfected and naïve cells directly and selectively indicates the involvement of this receptor to the measurement of any cell phenotype (cell death in our experiments).  Our long-term experience in screening compounds for neurotrophic activity, has demonstrated that molecules selected in this way have comparable receptor phosphorylation activity with natively receptor expressing cells. Additionally, the SH-SY5 line was chosen as a more straightforward alternative to other neuronal lines, for fast screening of the ability of compounds to induce neurite outgrowth, a hallmark of neural differentiation. SH-SY5Y are indeed commonly used to model neurons, demonstrating key features present in neurons such as constitutive activity of GPCRs (Ibrisimovic E, PMID: 22476939) and desensitization mechanisms (S J Briddon et al, PMID: 9831908) (Willets JM, PMID: 11856737). They have already been used also to study 5-HT4R-induced neuroprotection (Bianco F, PMID: 26893157).

We agree with the reviewer that both activities should be dissected further in subsequent in depth analysis of these short-listed candidates, to properly assess their potential as drug candidates in relative in vitro and/or in vivo systems. In line with that, we considered that assessing more complex aspects of their desirable drug-like properties, such as their BBB penetrability, were out of the scope of this initial purely in vitro screening study, since it demands the use of mouse models of disease, a time consuming experimental approach, which of course could provide a more comprehensive characterization of these compounds but significantly deviate from the initial intent and nature of the present work.

Experimental studies on permeability of the compound have unfortunately not been possible considering the short period provided for the review but an in silico prediction section has been added to the manuscript.

Minor revisions

The manuscript includes some typing errors and it should be re-checked.

Page 2, line 27, the sentence “the light of these results, we have “should be deleted.

The sentence is changed.

Page 6, line 145, compound LM22A-4 should be deleted in the caption of figure 3 since it is not included in the figure.

The caption is changed.

Page 7, line 172, “Ki” should be written as “Ki”. Please uniform the manuscript accordingly.

The change has been made.

Page 8, table 21, the concentration of compounds tested should be 1.10-8 and 1.10-6 M, please change the table header accordingly.

The table is changed.

Page 8, line 181, the sentence “behaves a partial agonist” should be changed in “behaves as partial agonist”.

The sentence is changed.

Page 11, line 268, the question mark should be deleted.

The question mark is deleted.

Page 17, line 558, “0.0625–2vnM” should be changed in “0.0625–2 nM”.

The sentence is changed.

Page 18, line 590, “2x106” should be changed in “2x106”.

The change has been made.

Overall the manuscript is well written and good English has been used, but some typos and careless errors need to be corrected.

English has been checked throughout the document.

Round 2

Reviewer 3 Report

Comments and Suggestions for Authors

The quality of the manuscript has been improved and in my opinion it can be published in the present form.